# Supplementary Nitric Oxide Donors and Exercise as Potential Means to Improve Vascular Health in People with Type 1 Diabetes: Yes to NO?

**DOI:** 10.3390/nu11071571

**Published:** 2019-07-12

**Authors:** Olivia McCarthy, Othmar Moser, Max L. Eckstein, Stephen C. Bain, Jason Pitt, Richard Bracken

**Affiliations:** 1Applied Sport, Technology, Exercise and Medicine Research Centre (A-STEM), College of Engineering, Swansea University, Swansea SA1 8EN, UK; 2Diabetes Research Group, Medical School, Swansea University, Swansea SA2 8QA, UK; 3Cardiovascular Diabetology Research Group, Division of Endocrinology and Diabetology, Department of Internal Medicine, Medical University of Graz, 8036 Graz, Austria

**Keywords:** type 1 diabetes, cardiovascular disease, nitric oxide, endothelial dysfunction, dietary nitrates

## Abstract

Type 1 diabetes (T1D) is associated with a greater occurrence of cardiovascular pathologies. Vascular dysfunction has been shown at the level of the endothelial layers and failure to maintain a continuous pool of circulating nitric oxide (NO) has been implicated in the progression of poor vascular health. Biochemically, NO can be produced via two distinct yet inter-related pathways that involve an upregulation in the enzymatic activity of nitric oxide synthase (NOS). These pathways can be split into an endogenous oxygen-dependent pathway i.e., the catabolism of the amino acid L-arginine to L-citrulline concurrently yielding NO in the process, and an exogenous oxygen-independent one i.e., the conversion of exogenous inorganic nitrate to nitrite and subsequently NO in a stepwise fashion. Although a body of research has explored the vascular responses to exercise and/or compounds known to stimulate NOS and subsequently NO production, there is little research applying these findings to individuals with T1D, for whom preventative strategies that alleviate or at least temper vascular pathologies are critical foci for long-term risk mitigation. This review addresses the proposed mechanisms responsible for vascular dysfunction, before exploring the potential mechanisms by which exercise, and two supplementary NO donors may provide vascular benefits in T1D.

## 1. Introduction

Type 1 diabetes (T1D) is an autoimmune disease characterised by the progressive depletion and destruction of pancreatic β-cells accompanied with impaired glucagon-producing α-cell function. The resulting deficiency in endogenous insulin secretion manifests in chronic hyperglycemia with the sequential need for a lifelong reliance on exogenous insulin therapy [1]. For individuals with T1D, cardiovascular disease (CVD) is a major cause of mortality and constitutes a major area of pharmacological and clinical research [2,3,4,5,6,7]. The term ‘CVD’ refers to an array of singularly distressing diseases that affect the micro-vascular and macro-vascular systems. Although the pathogenesis of these complications is multidimensional, the common recipient of injury is the vascular endothelium. This is a monolayer of cells that line the inner surface of the blood vessels. The single, adjacent formation of the endothelial cells (ECs) enables a tight confluent structure, which constitutes an interface between circulating blood and lymph in the lumen and the rest of the vessel wall. For many decades, the structural characteristic of the ECs meant the endothelium was viewed simply as a semipermeable barrier between blood and interstitial fluid, which facilitated the exchange of water and small molecules. However, more contemporary views highlight a mediatory role in an extensive range of vital homeostatic functions. Occupying a strategically important location between circulating blood and the surrounding tissues, the endothelium modulates the tone of the underlying vascular smooth muscle, maintains a non-adhesive luminal surface, mediates homeostasis, evokes cellular proliferation, and modulates inflammatory and immune mechanisms within the vascular wall [8]. Moreover, there is increasing evidence suggesting that neovascularization and vasculogenesis are products of increased proliferation of bone marrow–derived circulating l progenitor cells (cPCs) [9], which are reduced in people with T1D [10,11,12]. Endogenously, cPCs are upregulated in response to conditions of hypoxia, heat, stress, trauma, and altered energy status. Collectively, these conditions characterize the physiological responses to both exercise and nutrient intake, which can considerably alter bioenergetics.

EC activity can be dramatically altered in response to various physiological and pathophysiological stimuli. ECs that are actively undergoing proliferation exhibit phenotypical alterations in both their function and shape. Junctional structures contribute to the ‘resting’ phenotype by transducing signals within the cells and changing gene expression. For example, when the ECs are confluent, they express an epithelioid phenotype that inhibits growth and mobility [13]. The tight contact between each EC provides a form of protection against apoptosis. However, sparse cells, which lack cell-cell junctions, are unable to transduce such signals, which results in fibroblastic morphology, active growth, and cell mobility. The process is a tightly regulated balance between the intermediates that induce cell damage i.e., pro-constrictive, pro-inflammatory, pro-thrombotic, and pro-hypertensive factors and the intermediates that support cell vitality i.e., pro-relaxation, anti-inflammatory, anti-thrombotic, and anti-hypertensive factors. Disturbances to this fine-tuned equilibrium lead to ‘’endothelial dysfunction’’ (ED), which describes an overexposure of the ECs to repeated insults that compromise the integrity of the vessel wall [14]. 

Critically, the early manifestation of ED precedes intimal thickening and clinical atherosclerosis by decades. As such, the early presentation of ED not only increases the relative risk for a subsequent cardiovascular event but also constitutes a biological barometer for cardiovascular function. This is important considering the elevated risk of primary CVD in people with-versus without-T1D across the lifespan [15]. Taken collectively, the accelerated degree to which diabetes propagates atherosclerotic tendencies emphasises the need to target risk in its infancy before progression has reached an irreversible stage.

Research has emphasized the significance of nitric oxide (NO) bioavailability in the pathogenesis of ED and subsequent atherosclerosis [16,17]. Of particular interest are the vasodilatory effects of NO, which control the distensibility, compliance, and elastic modulus of the arterial-vascular system [18]. 

The compliance of the arterial networks can be accessed via flow mediated dilation (FMD), which detects changes in brachial artery diameter before and after ischemic stress induced by sphygmomanometer cuff inflation [1]. As an intimate reflection of NO bioavailability, a delayed or non-existent FMD response correlates with the extent of angiographically detectable vascular disorders [2] and the subsequent manifestation of cardiovascular events [3]. A reduced FMD response has been reported in those with-versus-without T1D and this phenomenon appears to present as early as childhood [4].

Therefore, maintaining a continuous pool of circulating NO becomes an increasingly important requirement in pathological circumstances characterised by low arterial compliance. Accordingly, interventions aimed at augmenting nitric oxide synthesis (NOS) via increases in enzymatic gene transcription, mRNA stability, and protein translation have generated interest within the literature. Furthermore, considering the identification of metabolic abnormalities in ED, the bioenergetic potential of acute and regular exercise alongside the contributory roles of macro-nutrient and micronutrient status should not be overlooked. As such, dietetic and nutritional interventions gated toward enhancing NOS constitute potential therapeutic options for promoting vascular health.

In this article, we first outline the hypothesized pathological basis of ED in T1D, before exploring existing literature that outlines the potential role of physical exercise and two dietary supplements acting as NO donors in preventing and/or managing vascular related complications. 

## 2. Nitric Oxide Synthesis and Biochemical Formation in The Vascular Endothelium

NO is a soluble gaseous signaling molecule formed in the vascular endothelium. Its formation is dependent on the NOS enzymes, which is a group of polypeptide proteins that consist of three genes including neuronal nitric oxide synthase (nNOS), inducible nitric oxide synthase (iNOS), and endothelial nitric oxide synthase (eNOS) [18]. NOS activity is regulated at transcriptional, translational, and post-translational levels, which differ depending on several genetic and environmental cues [18]. 

Both nNOS and eNOS are Calcium- Calmodulin (Ca^2+^- CaM) dependent enzymes while iNOS can operate independently of Ca^2+^ [19]. In order for the NOS proteins to actively synthesise NO, the two separate NOS enzymes must dimerise to produce a homodimer. The substrate for NOS enzymes is the amino acid L-arginine, which undergoes catabolism to L-citrulline concurrently producing NO in the process [18]. The first step involves the conversion of L-arginine into the molecule *N*-hydroxy-arginine via the reduction of nicotinamide adenine dinucleotide phosphate (NAPDH) to NADP^+^. The second step involves the addition of another NADPH molecule, which binds to the NADPH within the reductase domain. Lastly, the conversion of L-arginine into L-citrulline requires the addition of a half-reduced molecule of NADPH and an O_2_ molecule. During the cleaving process of *N*-hydroxy-arginine, the nitrogen atom is double bound to one of the oxygen (O_2_) molecules to create NO. The remaining O_2_ molecule along with the remaining hydrogen atoms are combined to create H_2_O (Figure 1).

The synthesised NO diffuses into circulating platelets and the smooth muscle cell layer where it activates guanylate cyclase (GCA) and then cyclic GMP (cGMP). The presence of cGMP initiates vascular relaxation and minimises platelet aggregation to provide equilibrium of pro-thrombotic and anti-thrombotic factors [20]. Collectively, this process is termed ‘endothelial-dependent vasodilation’ (EDV). However, while this endogenously regulated pathway is the most recognized for NO production, an alternative O_2_-independent pathway has been identified, during which conversion of inorganic nitrate (NO_3_^−^) to nitrite (NO_2_^−^) and then to NO occurs in a stepwise fashion (Figure 2). Following exogenous ingestion, NO_3_^−^ is sequentially taken up by the salivary glands before being reduced to NO_2_^−^ by oral commensal bacteria. The NO_2_^−^ in swallowed saliva is then internally absorbed and, once within the system, circulation is further reduced to NO by metalloproteins (haemoglobin and deoxyhaemoglobin), enzymes (xanthine oxidoreductase), and compounds with redox potential (polyphenols) [21,22]. Notably, the reductive process of NO_2_^−^ to NO is stimulated by low PO_2_ and pH, which perhaps emphasises the compensatory potential of the NO_3_^−^ reduction pathway in pathological complications that effect the efficiency of NOS activity including T1D.

## 3. Potential Pathogenic Mediators of Endothelial Damage in Type 1 Diabetes

By virtue of its position i.e., the interface between moving blood and tissue, the endothelium is exposed to a substantial amount of both biomechanical and biochemical stimuli. Pertinent to physical exercise, the biomechanical stimuli include wall shear stresses (tractive forces generated at the luminal endothelial interface by blood flow), pressures (hydrostatic forces that act perpendicular to the endothelial interface), and cyclic strains (circumferential stretching of endothelium cells within the vessel wall as a consequence of pulsatile blood flow) [23]. The biochemical stimuli include hyperglycemia, hyperlipidemia, and insulin resistance [14]. These stimuli can have both adaptive and maladaptive consequences and may dictate the overall phenotypic response of the ECs. 

### 3.1. Abnormal NO Production

While not fully understood, the hypothesized mechanisms responsible for the increased prevalence of ED in T1D is believed to cause EC apoptosis, NAPDH oxidase activation, eNOS uncoupling, and loss of NO bioavailability [24]. Paradoxically, research has shown a reduction in the suppressing activity of the NOS inhibitor asymmetric-dimethyl-L-arginine (ADMA) in people with short duration and uncomplicated T1D [25]. Mechanistically, the authors proposed that increased hyperglycemia-induced NOS by iNOS, along with a reduced inhibition of iNOS activity may lead to an over-production of NO [25]. Earlier work by the same group reported hypouricemia in people with T1D, which they hypothesized as another potential mechanism for the observed over production of NO [6]. While uric acid has both direct and indirect antioxidant properties, the latter phenotype acts by promoting superoxide dismutase activity. When the superoxide is overproduced together with NO, the resultant peroxynitrite subsequently oxidizes tetrahydrobiopterin, an iNOS, and eNOS co-factor, into hydrobiopterin. Under this condition, the iNOS and eNOS enzymes are in an uncoupled state. Thus, the increased serum uric acid may be a sign of ED, which is secondary to diminished vascular NO activity and the consequential lack of xanthine oxidase inhibition [26]. In a complete contract to its sister isoforms, the expression of iNOS is absent in regulatory physiology [7] and, yet, becomes apparent in conditions of pathophysiological inflammation [8,9]. Unsurprisingly, iNOS has been referred to as a contributary mediator of vascular distress [10,11] and may help explain the elevated rates of ED in those with T1D.

### 3.2. Hyperglycaemia 

Hyperglycemia is a feature of T1D and plays a major role in the susceptibility of vascular complications [27]. Biochemically, hyperglycemia enhances the activity of the four main gluco-mediated pathways, which are the polyol pathway [28,29], the hexosamine pathway [30], the formation of advanced glycation end-products (AGEs) [31], and the activation of protein kinase C (PKC) [32,33] where each orchestrate changes at the endothelial level. First, in the presence of high glucose, aldose reductase reduces glucose to sorbitol and, subsequently, fructose within the polyol-pathway. During this process, the aldose reductase concurrently consumes NADPH, which, thereby, increases the susceptibility of the cell to oxidative stress [29]. Second, the increased activation of the hexosamine pathway may be linked to a loss of insulin-mediated capillary recruitment via increases in glucosamine-6-phosphate. Notably, this results in the accumulation of *Ŋ*-acetylglucosamine and, subsequently, *O*-linked glycosylation (*O*-GlcNAc). Experimentally, *O*-GlcNAc has been shown to modify eNOS activity in T1D, such that the enzyme is rendered incapable of activation by fluid shear stress stimuli and vascular endothelial growth factor signaling [30]. Third, AGEs quench NO and impair the extracellular matrix and tissue remodeling [31], while glucose-induced changes in PKC isoform activity have been shown to reduce NO production [33].

At the cellular level, the ECs are particularly susceptible to hyperglycemia [34], which is believed to account for the observed impairments of vascular repair processes via augmented phosphorylation of eNOS and NO production [24]. In response to damage or injury, there is an upregulation in the release of ECs from the bone marrow niche, which respond by proliferating, migrating, and honing to the ischemic or damaged tissue. This process of repair and rejuvenation is largely based on a group of cPCs that have been shown to initiate vascular repair and regeneration [9]. Not surprisingly, several progenitor cell populations including circulating endothelial progenitor cells (cEPCs), mesenchymal stromal/stem cells (MSCs), and resident cardiac progenitor cells (CPCs) have stimulated interest as therapeutic targets in vascular pathologies [35]. The number and functionality of cPCs appear to be reduced in clinical T1D [10,11,12,36] which also manifests as a blunted cEPC response to acute exercise [37,38]. This blunted phenotype suggests that mobilization failure precedes the reduction of proangiogenic cell activity, which accelerates the likelihood of vascular distress. However, the combination of intensive insulin therapy alongside glucagon-like peptide-1 (GLP-1) as a means to optimize glycemia has been shown to restore endothelial function in people with T1D [39,40]. 

Moreover, coincubation with the NO donor sodium nitroprusside (SNP) or p38 mitogen-activated protein kinase (MAPK) inhibitor SB230580 have previously been shown to significantly ameliorate the inhibitory effect of hyperglycemia on the EPC number and proliferation (for early and late EPCs, respectively) [24]. However, coincubation with the NOS inhibitor l-Ng-nitro-l-arginine methyl ester (l-NAME) or PI3K inhibitor LY294002 has been shown to enhance the inhibitory effect of high glucose on the EPC number and proliferation [24]. 

Taken collectively, these data indicate that hyperglycemia may downregulate vascular repair mechanisms, and that the introduction of certain NO donors may inhibit or alleviate the magnitude of this response. Notwithstanding the mechanistic basis of these findings, research has consistently demonstrated that improvements in glycemic control translate into improvements in both clinical [41,42,43] and subclinical [44] biomarkers of CVD in individuals with T1D. 

## 4. The Energetic and Therapeutic Potential of Exercise-Induced Nitric Oxide Synthase

With the onset of physical exercise, Ca^2+^ travels into the cytoplasmic portion of the skeletal muscle cell, where it can transform chemical energy into mechanical work through the cyclic interaction between actin and myosin filaments [45]. During exercise, cytosolic and mitochondrial Ca^2+^ levels increase exponentially alongside intensity dependent changes in metabolic contributions [46,47]. These extracellular signals facilitate the interaction of the NOS enzymes with CaM to ultimately generate NO [18]. The importance of NO in modulating skeletal muscle function is strongly inferred by the presence of all three NOS isoforms in mammalian myocytes [18]. Skeletal muscle functions regulated by NO include contractility, autoregulation of blood flow, nutrient exchange, and mitochondrial respiration [18]. Although all isoforms are transcriptionally regulated by hypoxia, the eNOS isoform found in vascular and skeletal muscle cells is upregulated in response to physical exercise mediated via both biomechanical (hyperemia) and biochemical (metabolic and extracellular) processes [18]. The involvement of NO in mediating the autoregulation of blood flow during the excitation - contraction process, hypoxia, and the reflex sympathetic discharge experienced in resting and exercising skeletal muscle highlights its significance in determining the microvascular responses to exercise [18]. The co-operative intimacy between skeletal muscle and its surrounding vasculature is made clear by the proximity through which muscle fibers and capillaries are found [32]. The rate of tissue perfusion is dependent on the expansion of the arteriole lumen, which dictates the amount of blood perfused into the capillary networks for subsequent venous return. The elastic potential of the large vessels within the arterial system produce pressure forces that facilitate the movement of oxygenated blood into the smaller vessels via elastic recoil [48]. This mechanism pushes the blood into the arterioles and subsequently the capillary system, which characterizes a network of small tubular structures that facilitate nutrient exchange. During exercise, there is an increased need for O_2_ by the peripheral musculature, which is physiologically met by the sudden rise in arterial blood flow [48]. Concurrently, there is an acute increase in intraluminal shear forces that induce EDV in order to facilitate nutrient exchange and local muscle capillary-and-arterial vasodilatation [49]. Structurally, muscle microvasculature is the final interface through which circulating nutrients, hormones, gases, and electrolytes must pass in journeying to and from the systemic circulation [50]. Markedly, a decrease in capillary density also effects the spatial pattern of flow within the microvascular beds, which propagates non-uniformity in the distribution of vessel flow and, therefore, tissue nutrient exchange [32]. 

With this in mind, microvascular dysfunction, in particular rarefaction, may have consequences for skeletal muscle perfusion during exercise, due to diminished nutrient delivery to the contracting myocytes [32]. Supporting this concept, research has shown impaired peak and submaximal physiological parameters during cardiopulmonary exercise testing in individuals with T1D [44,51,52]. These dysfunctional mechanisms correlate with low flow mediated dilatation of the brachial artery and a blunted response to hyperemia if retinopathy is present, which highlights the downstream complications associated with vascular dysfunction [53]. It should be noted that the attainment of good glycaemic control, that is a HbA_1c_ of at least < 7% (53 mmol/mol), appeared to offset this observation [54]. Yet, only 27% of adults with T1D are currently achieving this glycemic target [55]. Perhaps more worryingly, several studies have demonstrated suboptimal glycemic control in pediatric and adolescent T1D cohorts across various nationalities [56,57,58], which may further exacerbate vascular disease susceptibility [59]. The fact that the initiation of CVD precedes the manifestation of clinically recognized biomarkers, such that atherosclerotic tendencies appear as early as childhood irrespective of diabetes [60], combined with the increase in T1D diagnosis in adolescents within the last decade [61], emphasizes the need to target risk in its infancy. Considering the intimate link between T1D and microvascular dysfunction [5], the potency of physical exercise to increase skeletal muscle fiber capillarisation and facilitate efficient gas and nutrient exchange, should lessen the likelihood of vascular distress. 

During the last decade, several research groups have demonstrated that both acute and chronic exercise training has the potency to initiate vascular repair via mechanisms of re-endothelization [62,63,64,65,66,67]. Thus far, research has demonstrated a blunted cEPC response to physical exercise in people with T1D, which reflects the above hypotheses regarding maladaptation’s at the microvascular level. However, it should be noted that, in these instances, the importance of global conditioning has been somewhat neglected, since the exercise programming has consisted of either exclusively moderate-intensity continuous exercise [38] and/or submaximal, hypertrophic lower limb resistance exercise [37]. Rather, it seems that ischemic, exhaustive exercise appears to be the most potent stimuli for increasing stem/progenitor cell activity within the bone marrow [68,69] and peripheral blood [69,70,71] albeit it in non-T1D cohorts. Supporting these insights, recent work by Boff and colleagues highlighted the superior effects of high intensity interval training versus moderate continuous exercise training on flow mediated endothelial function and cardiorespiratory fitness (CRF) in complication-free people with T1D [72]. 

Biochemically, intense exercise constitutes a metabolic insult, which causes considerable disturbances in various cellular and systematic tissue level processes. These metabolic disturbances activate several kinases and phosphatases, which are necessary not only for the immediate supply of energy to sustain contractional output, but also the synthesis of genetic transcriptions, that produce an adaptive phenotype for subsequent functional demands [32]. Intense exercise protocols have been shown to augment the production and release of several vascular repair-and-pro-angiogenic factors [64,73,74,75]. These processes are mediated via an upregulation in several intracellular signaling cascades including both the phosphatidyl inositol 3-kinase (PI3K) and AMP-activated protein kinase (AMPK) pathways [46,76], which further enhance glucose uptake and glycogen synthesis [77]. Recent research has identified a unique gene expression profile in the T1D heart and kidney ECs, which present opposite metabolic cues and distinct angiogenic patterns. The findings revealed an upregulation in several genes that inhibit angiogenic, tissue remodeling, and cell differentiation processes as well as an overexpression of AMPK-associated genes implicated in catabolism in renal ECs coupled with a downregulation of these same genes in heart ECs [78]. The discovery that NOS is also a downstream target of AMPK in heart cells raises the possibility that NO could reinforce the intracellular activity of this enzyme via an auto-stimulatory loop [77] as well as highlights the therapeutic potential of a mediator of AMPK activity. These data suggest that, in pathologies during which there is an alteration in metabolic proficiency, supplementary NO donors alongside exercise may represent a compelling solution to the often-blunted responses experienced, possibly by means of offering a supplementary ‘boost’ in the functionality of comprised physiological processes. Despite these insights, the realms of both appropriately designed exercise programs combined with potential ergogenic and vasodilatory supplements remains relatively unexplored in individuals with T1D, who may stand to benefit most from interventions that acutely enhance vascular performance.

## 5. Potential Therapeutic Mediators of Endothelial Vitality in Type 1 Diabetes 

The complexity of ED within T1D requires a multimodal approach that involves standardized care alongside reference to modifications in behavioral and environmental factors. In hypoxic conditions, when the O_2_-dependent NO synthases may become dysfunctional, the nitrite reduction process is instead greatly enhanced. As such, the utility of nitrate and nitrite as storage pools supporting NO signaling during metabolic stress is of interest [21]. With new gene-based research having identified distinct metabolic and angiogenic patterns in T1D ECs, the role of nutrient status in controlling the activity of various metabolic pathways is noteworthy. Certainly, research orientated around the exploration of nutrient-gene interaction and expression has attracted more attention, with nutrigenomics gaining more momentum in the scientific community. As such, dietary and/or exercise practices that may enhance NO constitute potential therapeutic options that warrant further exploration.

### 5.1. NO Donor 1: Dietary Nitrate 

Experimentally, it has been demonstrated that plasma NO_2_^−^ mirrors acute changes in eNOS activity, such that low circulating levels correlate with the number of CVD risk factors [79,80] Unsurprisingly, NO_2_^−^ has been identified as an important mediator of several cardiovascular-based functions. 

This provides a rationale to intervene with substances rich in NO_3_^−^ (with the sequential increase in circulating NO_2_^−^) as a potential means of modulating endothelial health and CVD susceptibility. Beetroot juice (BRJ) is one of the richest dietary sources of NO_3_^−^ noted for its high degree of intestinal bioavailability [81]. Beetroot (Beta vulgaris) is classified as an herbaceous biennial from the chenopodiaceous family that has several varieties made distinctive by differences in their taste and taproot color [82]. The deep red colored beetroots contain several active compounds (carotenoids, glycine betaine, saponins, betacyanin’s, folates, betanin, polyphenols, and flavonoids) [82], which account for its micronutrient profile. Notably, these chemical compounds appear to assist in elevating circulating levels of NO_2_^−^ and, therefore, harness the potential to improve vascular compliance [83,84,85] Recently, a large collection of literature has demonstrated the efficacy of BRJ in elevating levels of plasma NO_2_^−^ (and, therefore, the potential for O_2_ independent NO production) to increase skeletal muscle blood flow, slow the reduction in microvascular O_2_ partial pressure (PO_2_), reduce O_2_ uptake (VO_2_) [86,87,88], lower exercise mean arterial oxygen saturation (Sa O_2_) [89], and improve cognitive performance during exercise [87,90]. Moreover, these performance enhancing effects have been observed across multiple exercise disciplines (water sports [89,91,92], running [86,90,93], cycling [87], and strength-based activities [94,95] performed at varying intensities (high [86,90,91,93], moderate [88,92], and low [89,95]) and within several contrasting environments (below sea level [89], hyper-thermic conditions [96], and altitude [88]). 

Despite these findings, research has so far failed to identify any improvements in either blood pressure or exercise performance outcomes following four days dosing of BRJ (6.43 mmol·L^−1^ NO_3_^−^) in people with T1D. Moreover, while a post-exercise improvement in peripheral indices of vascular function was noted, this did not translate to elevations in cellular indices of vascular function, since a blunted EPC response was observed [37]. However, it should be noted that the exercise protocols used have somewhat neglected the implementation of global conditioning (i.e., the practice of simultaneously engaging a large amount of muscle mass in both the upper and lower body segments to induce substantial tissue and metabolic disturbance), which appears to be a critical component in provoking a response [64]. Physiologically, it is proposed that conditions of cellular hypoxia stimulate erythrocyte-derived NO-mediated vascular relaxation, which serves to match skeletal muscle oxidative capacity (Q.O_2_) to a local metabolic rate, while high PO_2_ triggers vasoconstriction [97]. 

Reduced O_2_ levels in skeletal muscle during intense exercise mandate the utilization of non-oxidative fuel metabolism that contributes to conditions of hypoxia. Recent research has supported these concepts, which shows a higher degree of PO_2_ in muscle comprised of fast twitch rather than slow twitch fibers following NO_3_^−^ supplementation. These findings corroborate the hypothesis that the physiological environment extant in these muscles (i.e., lower pH and higher lactate) appears favorable for the reduction of NO_2_^−^ to NO [98]. Furthermore, it is postulated that the activity of the NOS family of enzymes may be reduced under such conditions. Reference [21] allows the NO_3_^−^NO_2_^−^-NO pathway to serve as a backup system in the local regulation of NO bioavailability when the endogenous L-arginine pathway is dysfunctional [21,99]. Dietary NO_3_^−^ supplementation also abolishes the reduction in the rate of pH-independent phosphocreatine (PCr) recovery, which is typically observed in hypoxia and restores maximal oxidative capacity (Q._max_) to values similar to those observed in normoxia [97]. A significantly lower muscle pH at rest and at the end of exercise has been observed in people with T1D, which indicates a greater reliance on glycolytic metabolism [100]. Moreover, a significantly slower PCr recovery time has been noted in adolescence with-versus-without T1D, which suggests a reduced skeletal muscle oxidative profile with an impaired recovery capacity [101]. 

Taken collectively, these data suggest that, during hypoxia, dietary NO_3_^−^ supplementation may facilitate NO production and enable greater muscle oxygenation [97], which offsets the possible manifestation of hypoxia induced muscle fatigue (which is exacerbated in those with T1D [102]). Furthermore, the implementation of whole-body exercises that require the simultaneous recruitment of large amounts of skeletal muscle in both the upper and lower body segments also appear to benefit most from NO_3_^−^ supplementation. During whole body exercise in hypoxia, NO_3_^−^ supplementation has been shown to reduce the O_2_ cost of submaximal exercise [103,104] as well as elevate muscle oxygenation [104]. Critically, research has demonstrated abnormal peripheral skeletal muscle perfusion both at rest and after ischemia in individuals with T1D, which also appears to correlate with impaired endothelial function and reactive hyperemia when retinopathy is present [32]. During exercise, this under perfusion (which potentially effects both the skeletal and cardiac muscle tissues) may lead to peak performance limitations especially considering the lowered nutrient delivery capabilities associated with hyperglycemia-induced hexosamine activity. Several studies have created an upregulation in NO by means of the supplementary inorganic NO_2_^−^, which appears to have positive exercise tolerance outcomes in subjects with and without vascular dysfunction [105]. Considering the strong inverse and independent association between acute non-fatal CV events and cardio-respiratory fitness (CRF), the ergogenic potential of BRJ has important health and performance related outcomes.

### 5.2. NO Donor 2: Citrulline Malate

L-citrulline is one of the three amino acids involved in the urea cycle (alongside L-arginine and L-ornithine). The cycle is divided into two main parts including a chemical part that occurs within hepatocytes and operates to form urea from ammonia (NH_3_), and a mechanical part that occurs within the kidneys and operates as a filtration system with the potential to synthesize NOS. Amino acids undergo deamination (the removal of ammonium [NH_4_]) and form keto-acids, which undergo decarboxylation i.e., the liberation of carbon dioxide (CO_2_) and produce adenosine triphosphate (ATP). The NH_4_^+^ is then transported into the hepatocyte mitochondria concurrent with the entry of carbon dioxide (CO_2_) produced via bicarbonate (HCO_3_^−^) [106]. Upon their entry, NH_4_^+^ and CO_2_ are acted on by the enzyme carbamoyl phosphate synthetase one (CPS1). CPS is synthesised via *N*-acetylglutamate, which is produced by Acetyl-CoA plus glutamate via the enzyme *N*-acetylglutamate synthase. The *N*-acetylglutamate synthase enzyme is regulated by the presence of arginine, which emphasises the importance of substrates like L-citrulline that facilitate an upregulation in circulating concentrations of L-arginine [106]. Research has demonstrated that the availability of L-citrulline facilitates the clearance of ammonium, which is an upstream mediator of glycolysis and inhibitor of oxidative metabolism that contributes to excessive lactate formation [107]. While ureagenesis is responsible for a small percentage of L-citrulline synthesis, the majority of circulating citrulline is converted to L-arginine via the hydroxy-L-arginine and NOS pathway described previously. 

While supplemental arginine can be employed to enhance intracellular arginine/ADMA ratios, L-arginine treatment is retarded by intestinal arginase activity, which results in considerable pre-systemic elimination [108]. Paradoxically, although endogenous L-arginine is the direct precursor of NO production, L-citrulline supplementation has been shown to raise circulating L-arginine concentrations to a greater extent than L-arginine supplementation during endotoxemia and mitochondrial dysfunction [109,110,111]. As such, oral supplementation with L-citrulline may appear to be a preferred choice in pathologies associated with elevated levels of inflammation [111,112].

In a diabetic nephropathy-induced rodent model, researchers found that L-citrulline but not L-arginine was effective in preventing pathologically-induced increases in the glomerular filtration rate and proteinuria [113]. More recently, the chronic co-administration of L-citrulline and sepiaterin (a biosynthetic precursor of tetrahydrobiopterin i.e., an important co-factor of eNOS coupling [114]), had a favorable impact on the evolution of diabetic cardiomyopathy and myocardial ischemia/reperfusion injury in obese diabetic mice [115]. Additional work has demonstrated the efficacy of L-citrulline supplementation in improving indices of endothelial function from ADMA-induced injury [116,117]. Moreover, improvements in the right ventricle function in heart failure patients with preserved ejection fraction have been noted following supplementary L-citrulline [118]. Pooled analysis has also highlighted the anti-hypertensive properties of L-citrulline, which were most apparent at doses of ≥6 grams per day [119]. However, it should be noted that these vascular improvements are not ambiguous within the literature, with some research failing to identify flow mediated improvements with acute and short-term L-citrulline administration [120,121].

Malate is one of the eight intermediates of the citric acid cycle (CAC) alongside citrate, iso-citrate, alpha-ketoglutarate, succinate, succinyl-CoA, fumarate, and oxaloacetate. The intermediates operate in a cyclic sequence, which is why an addition of any one intermediate to the cycle has an anaplerotic effect, while its removal has a cataplerotic effect. During each cycle, these anaplerotic and cataplerotic reactions operates to remove electrons from acetyl CoA for the formation of Nicotinamide adenine dinucleotide (NADH) and flavin adenine dinucleotide (FADH_2_). The re-oxidation of these electrons during oxidative phosphorylation enter the electron transport chain where they are used to generate ATP. Thus, as one of the gluconeogenic intermediates produced by the CAC, malate also plays an integral role in the process of gluconeogenesis by facilitating the reduction of pyruvate to oxaloacetate within the mitochondria. Pyruvate must first be translocated into the mitochondria where it can be acted on by pyruvate carboxylase in the presence of acetyl CoA and Biotin (Vitamin. B7) to form oxaloacetate. However, the oxaloacetate is incapable of transporting out of the mitochondria matrix. Therefore, it must first be reduced to form malate, which can transverse the mitochondrial matrix via the malate aspart shuttle. Once converted to malate, the malate aspartate shuttle enables its navigation across the anti-porter transport system housed on the inner membrane of the mitochondria matrix and into the cytosol in exchange of an α keto-glutamate molecule. After its deportation from the mitochondrion, malate is converted into cytosolic oxaloacetate, which is decarboxylated to phosphoenolpyruvate by phosphoenolpyruvate carboxykinase (Figure 3). This constitutes the rate limiting step in the conversion of nearly all the gluconeogenic precursors (such as the glucogenic amino acids and lactate) into glucose by the liver and kidney [122].

Combined, the synergistic use of L-citrulline and malate to produce citrulline-malate (CM), has generated interest as a potential therapeutic aid against vascular distress. Recent research has demonstrated improvements in mean arterial pulmonary hypertension (APH) and indices of quality of life in patients with idiopathic APH and Eisenmenger syndrome [123]. Earlier animal work by Callis et al. found that CM increased hepatic ureagenesis and favorized the renal reabsorption of bicarbonates, which offsets acidosis [124]. Moreover, as a commercially available sports supplement, CM has been linked to positive exercise performance outcomes, which are mainly driven by its anti-asthenic effect on muscle fatigue [107,125,126,127,128]. However, there is considerable disagreement on this topic within the literature, since several studies have failed to observe any ergogenic potential of CM regardless of variations in dosing strategies [129,130,131,132].

Nevertheless, considering that individuals with T1D report lower levels of CRF [51], which alone stand as both a major barrier to regular exercise engagement [133] and a biomarker of increased CVD. The ergogenic potential of CM as a mediator of continued exercise performance is noteworthy. Notwithstanding the importance in optimizing CRF from an exercise performance point of view, the necessary improvements in the functionality of the cardio-respiratory, vascular, and muscular parameters needed to achieve enhanced exercise tolerance contribute to minimizing CVD susceptibility, which is mechanistically underpinned by the integrative function of these systems.

## 6. Directions for Future Research

Despite an encouraging body of work having been done in individuals without T1D, the realms of ergogenic and vasodilatory dietary supplements remain relatively unexplored in individuals with T1D, who may stand to benefit most from interventions that acutely enhance vascular performance. Furthermore, while the acute endothelial responses to exercise have not been the main topic of discussion in this review, the potential of dietary supplements in combination with appropriately programmed exercise regimes that specifically target the upstream mediators of endothelial mediated vasodilation represent exciting exploratory opportunities, particularly considering the dampened endothelial response to acute exercise observed in individuals with T1D [37,38].

## 7. Conclusions

Intuitively, it seems reasonable to suggest that an increase in circulating NO_2_^−^ may be a means of alleviating the risk of diabetes-related vascular complications in people with T1D. However, while the biochemical mechanisms support the potential of these concepts in people with T1D, more research is needed to confirm their real-world applicability. 

## Figures and Tables

**Figure 1 nutrients-11-01571-f001:**
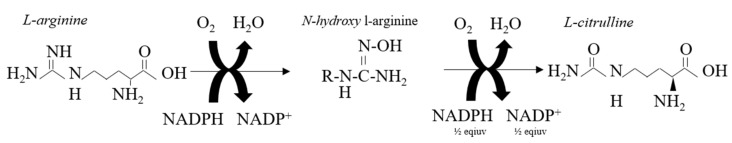
Biochemical formation of nitric oxide in the endothelium. NADPH = Nicotinamide adenine dinucleotide binding domain. NO = nitric oxide. O_2_ = Oxygen. Equiv = Equivalent. H = hydrogen. C = Carbon. N= Nitrogen. Modified from McAllister & Laughlin 2006 [5]. ‘Vascular nitric oxide: effects of physical activity, importance for health’. Essays Biochemistry vol 42. Figure 2. https://www.ncbi.nlm.nih.gov/pubmed/17144884.

**Figure 2 nutrients-11-01571-f002:**
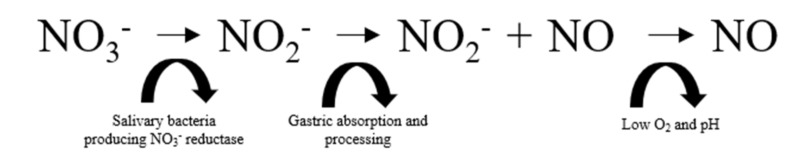
Exogenously regulated NO production pathway that involves the stepwise conversion of inorganic nitrate to nitrite and subsequently nitric oxide. NO_3_^−^ = inorganic nitrate. NO_2_^−^ = nitrite. NO = nitric oxide. O_2_ = Oxygen.

**Figure 3 nutrients-11-01571-f003:**
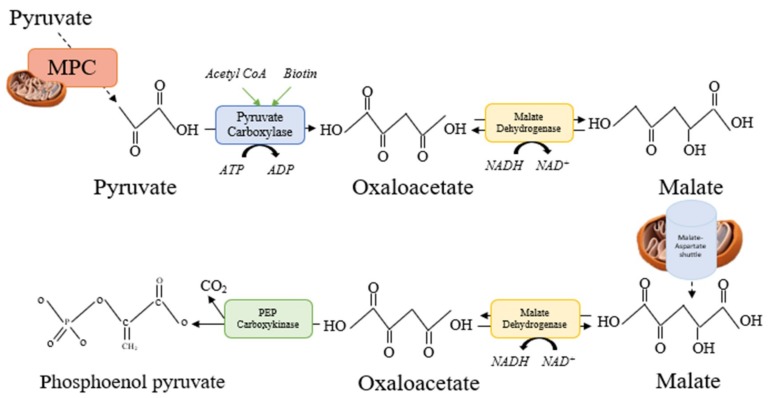
Graphical overview of the role of Malate in part of the gluconeogenic process. NADPH = Nicotinamide adenine dinucleotide binding domain. H = Hydrogen. O = Oxygen. MPC = mitochondrial pyruvate carrier. ATP = Adenosine triphosphate. ADP = Adenosine diphosphate. C = carbon.

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
