# Peer review of "Supplementary Nitric Oxide Donors and Exercise as Potential Means to Improve Vascular Health in People with Type 1 Diabetes: Yes to NO?"

_nutrients, 2019, doi:10.3390/nu11071571_

Round 1

Reviewer 1 Report

This manuscript is a review that focuses on the role of NO in the mechanisms and prevention of vascular disease in type 1 diabetes.

It also discusses the effects of exercise and dietary nutrients and supplements as improvement of peripheral NO.

Criticizes

1.It is unclear whether this review is limited to type 1 diabetes rather than type 2 diabetes.

2.Although it is clear that NO is associated with vascular disorders, there seems to be no evidence that the enhancement of peripheral NO production or the replacement of NO improves vascular disorders.

3.In paragraph Line 147, "Abnormal NO production", it is insufficient to explain what happens regardless of hyperglycemia.

4.FMD (Flow Mediated Dilation)is a test that examines the vascular endothelial function and appears to be associated with NO production ability. We think that we should discuss FMD.

Author Response

Point 1: It is unclear whether this review is limited to type 1 diabetes rather than type 2 diabetes.

Response 1: Dear reviewer, thank you for your comment and apologies for any confusion caused. The manuscript relates only to individuals with type 1 diabetes but due to a lack of literature within the type 1 diabetes sphere, we have referenced other population groups when necessary. Where possible we have clarified the studies that have used individuals with type 1 diabetes with reference to this throughout the text.

Point 2: Although it is clear that NO is associated with vascular disorders, there seems to be no evidence that the enhancement of peripheral NO production or the replacement of NO improves vascular disorders.

Response 2: Thank you for your comment. In this review, we put forward a model that discusses the potential therapeutic benefit of NO donors and exercise as means to reduce the likelihood of adverse events occurring in people with T1D. This review details the physiological importance of NO in vascular vitality with reference to the physiological mechanisms supporting this.

Point 3: In paragraph Line 147, "Abnormal NO production", it is insufficient to explain what happens regardless of hyperglycemia.

Response 3: Thank you for your comment.  This review is centred around the plausibility of exercise and NO donors to alleviate vascular complications specifically in individuals with T1D. We therefore felt it best to focus on the metabolic abnormalities pertinent to T1D i.e. hyperglycaemia as the main point of reference in the pathogenic mediators of endothelial damage section. The section has been amended to provide clarity and includes supporting references of the statements made. Please see lines 149-166.

Point 4: FMD (Flow Mediated Dilation)is a test that examines the vascular endothelial function and appears to be associated with NO production ability. We think that we should discuss FMD.

Response 4: Thank you for your comment. The purpose of this review is focused on the potential of NO donors and/or exercise as potential means to alleviate vascular distress in those with T1D with a particular emphasis on the cellular-endothelial responses. However, in light of your suggestion, a section on FMD has now been added with reference to its utility in detecting impairments in vascular compliance.  Please see lines 80-86.

Reviewer 2 Report

The manuscript was very well written.  It provided a tight review on the topic of where we stand regarding not only our understanding of nitric oxide for vascular health, but specifically for T1D.  The exercise and supplement sections are well handled.   

There is one section that merits consideration as a side box or appendix piece —the malate biochemistry starting line 398.  It takes the discussion too far from the topic.   

Author Response

Point 1: The manuscript was very well written.  It provided a tight review on the topic of where we stand regarding not only our understanding of nitric oxide for vascular health, but specifically for T1D.  The exercise and supplement sections are well handled.   

Response 1: Dear reviewer, thank you kindly for your comments and support of the manuscript draft.

Point 2: There is one section that merits consideration as a side box or appendix piece —the malate biochemistry starting line 398.  It takes the discussion too far from the topic.   

Response 2: Dear reviewer, thank you for this suggestion. Please note that the section has been amended to scale down the level of in-depth biochemistry included.

Round 2

Reviewer 1 Report

This manuscript is a review of the role of NO in vascular complications in type 1 diabetes and the potential for therapeutic intervention. There is no additional criticize.